# Unraveling the Polysaccharide Biosynthesis Potential of *Ganoderma lucidum*: A Chromosome-Level Assembly Using Hi-C Sequencing

**DOI:** 10.3390/jof9101020

**Published:** 2023-10-16

**Authors:** Ignat V. Sonets, Nikita V. Dovidchenko, Sergey V. Ulianov, Maria S. Yarina, Stanislav I. Koshechkin, Sergey V. Razin, Larissa M. Krasnopolskaya, Alexander V. Tyakht

**Affiliations:** 1Institute of Gene Biology, 34/5 Vavilova Street, 119334 Moscow, Russia; ignatsonets@gmail.com (I.V.S.); sergey.v.ulyanov@gmail.com (S.V.U.); sergey.v.razin@inbox.ru (S.V.R.); a.tyakht@gmail.com (A.V.T.); 2Knomics LLC, 34 Bld. 1 Narodnogo Opolcheniya Street, 123423 Moscow, Russia; tierprot@gmail.com (N.V.D.); st.koshechkin@knomx.com (S.I.K.); 3Institute of Protein Research, 4 Institutskaya Street, 142290 Pushchino, Russia; 4Faculty of Biology, Lomonosov Moscow State University, GSP-1, Leninskie Gory, 119991 Moscow, Russia; 5Gause Institute of New Antibiotics, 11 B. Pirogovskaya Street, 119021 Moscow, Russia; maria.s.yarina@gmail.com; 6Center for Precision Genome Editing and Genetic Technologies for Biomedicine, Institute of Gene Biology, 34/5 Vavilova Street, 119334 Moscow, Russia

**Keywords:** Hi-C-assisted genome assembly, medicinal mushrooms, chromosome conformation capture, fungal polysaccharides

## Abstract

*Ganoderma lucidum* exhibits the ability to synthesize a diverse range of biologically active molecules with significant pharmaceutical potential, including xylomannan and fucogalactan, which have demonstrated antitumor activity. However, there exists considerable intra-species variability in the capacity to produce these metabolites at high concentrations, likely reflecting the high genomic diversity observed from a limited number of strains sequenced to date. We employed high-throughput shotgun sequencing to obtain the complete genome sequence of *G. lucidum* strain 5.1, which is distinguished by its remarkable xylomannan synthesis capabilities. Through the utilization of semi-automatic reordering based on conformation capture (Hi-C) data, we substantially enhanced the assembly process, resulting in the generation of 12 chromosome-level scaffolds with a cumulative length of 39 Mbp. By employing both de novo and homology-based approaches, we performed comprehensive annotation of the genome, thereby identifying a diverse repertoire of genes likely involved in polysaccharide biosynthesis. The genome sequence generated in this study serves as a valuable resource for elucidating the molecular mechanisms underlying the medicinal potential of *Ganoderma* species, discovering novel pharmaceutically valuable compounds, and elucidating the ecological mechanisms of the species. Furthermore, the chromosome contact map obtained for the first time for this species extends our understanding of 3D fungal genomics and provides insights into the functional and structural organization within the fungal kingdom.

## 1. Introduction

Basidiomycetes comprise a substantial phylum, Basidiomycota, within the Kingdom Fungi, encompassing more than 40,000 species [1]. These fungi exhibit remarkable diversity in their metabolites, both structurally and functionally. Many of these compounds are exclusive to basidiomycetes and are not found in other fungal taxa or bacteria. This metabolic diversity arises from the utilization of shorter biosynthetic pathways compared to ascomycetes, implying that basidiomycetes have evolved advantages in maintaining the chemical and functional diversity of metabolites at a lower cost [2].

*Ganoderma lucidum* (Curtis) Karst *sensu lato* is widely recognized for its significant production of biologically active metabolites and stands out as one of the most prominent examples. In its natural habitat, this widespread species induces white rot in wood and is predominantly found in tropical and subtropical regions. With a history spanning over 4000 years, *G. lucidum* has been extensively employed in Asian traditional medicine to enhance health, vitality, and longevity. Decades of scientific exploration have led to the isolation, structural elucidation, and characterization of numerous metabolites derived from *G. lucidum*. These biologically active metabolites encompass a wide array of chemical classes and exhibit diverse biological functions. Among them, polysaccharides, polysaccharide-peptide complexes, triterpenes, phenols, proteins, peptides, steroids, nucleotides, and lectins have been identified, showcasing immunomodulatory, antitumor, antioxidant, antibacterial, antifungal, antiallergic, hepatoprotective, hypoglycemic, hypolipidemic, and other properties. Notably, polysaccharides and triterpenes demonstrate a broad spectrum of biological effects and exhibit high activity [3,4,5,6,7,8].

*G. lucidum* polysaccharides have undergone extensive investigation in both in vitro and in vivo experiments, revealing numerous pharmacological effects. These effects include anticancer properties [8,9], immunomodulatory [10], antiviral [11], antidiabetic [12], anti-inflammatory [13], hypolipidemic [14], gastroprotective [15], hepatoprotective effects [16], neuroactive [17,18], wound healing [19] and antioxidant capacities [20]. These biological activities of *G. lucidum* polysaccharides are attributed to their presence in various parts of the fungus, including the mycelium, fruiting bodies, basidiospores, and culture broth. The specific composition of monosaccharides, types of glucosidic bonds, tertiary structure, branching, and molecular weight play crucial roles in determining the polysaccharides’ biological activity [4,7]. *G. lucidum* polysaccharides can be categorized as water-soluble or alkali-soluble compounds, with water-soluble polysaccharides receiving relatively more extensive research attention compared to alkali-soluble ones [21,22]. The biologically active polysaccharides encompass both homopolysaccharides, primarily β-D-glucans, and heteropolysaccharides [4,13,23].

Accurate taxonomic identification of the producer strain is essential for evaluating the biosynthetic potential of the species and, ultimately, for documenting its pharmaceutically relevant metabolites. However, despite advancements in molecular approaches, the taxonomy of the genus *Ganoderma* and the *G. lucidum* complex remains highly contentious [24]. A critical task involves distinguishing between *G. lucidum sensu stricto* and other laccate species within the genus *Ganoderma* [25,26,27]. Studies in this field have revealed that the cultures of the genus *Ganoderma* used in China for therapeutic and prophylactic product production largely do not belong to the species *G. lucidum sensu stricto* but instead belong to the species *G. lingzhi*. Molecular investigations have confirmed that the former has a native geographic distribution in Europe and certain regions of China, while the latter is indigenous to East Asia [25].

Molecular phylogenetic analysis of our collection of *G. lucidum sensu lato* strains based on ITS rDNA sequences showed that strain 5.1 is *G. lucidum sensu stricto* [28]. Previously, two new polysaccharides were isolated from the submerged mycelium of the strain *G. lucidum* 5.1—alkali-soluble highly branched xylomannan and water-soluble fucogalactan—and their structures were established. The content of xylomannan in submerged mycelium was 2.6%, and fucogalactan was 0.1% [21,29]. In vivo experiments using murine models with transplanted tumors have demonstrated the significant antitumor activity of both xylomannan and fucogalactan, with no observed toxic effects. Notably, the highest rates of tumor growth inhibition were observed in animals treated with xylomannan [30]. The main chain of xylomannan is composed of (1-3)-linked α-D-mannopyranose residues, with a significant portion of them being substituted at position 4 by either single residues of β-D-xylopyranose or disaccharide residues of β-D-Manp-(1→3)-β-D-Xylp-(1→ or similar structures. Fucogalactan, on the other hand, is composed of galactose and fucose residues in a 4:1 ratio. Its molecule mainly consists of linear chains of (1→6)-linked α-D-galactopyranose residues, with every fourth residue of the main chain at position two carrying a side branch in the form of an α-L-fucopyranose residue. Both polysaccharides exhibit branching and may undergo non-covalent modifications during synthesis, such as the attachment of fatty acids. Synthesis of these polysaccharides in various fungi may involve Och1p, Van1p, Anp1p, Mnn9p, Mnn2p, and Mnn6p enzymes [31].

The antitumor properties of xylomannan are attributed to its ability to activate cellular and humoral immunity. Xylomannan has been shown to activate human peripheral blood mononuclear cells obtained from 22 donors. It induced the production of TNFα, a key cytokine involved in antitumor immunity. Furthermore, in most cultures of mononuclear cells stimulated by xylomannan, an increase in the production of IL-1β, IL-6, IL-8, and IL-10 has been observed. The stimulatory effect of xylomannan was comparable to that of phytohemagglutinin, a known immune stimulant. However, xylomannan did not result in an increase in IFNγ production [32]. In contrast to xylomannan, the total fraction of water-soluble polysaccharides isolated from the mycelium of *G. lucidum* strain 5.1 actively induced IFNγ in human blood leukocyte cultures [33]. The total fraction of water-soluble endo-polysaccharides and the ethanol-soluble low-molecular-weight compounds from the strain 5.1 mycelium exhibited a high chelating capacity comparable to that of EDTA-Na2 (used as a positive control). Additionally, the ethanol extract of the mycelium demonstrated antiradical activity against DPPH and inhibited oleic acid peroxidation [34].

The characterization of *Ganoderma* genomes holds immense potential for unraveling the mechanisms involved in the synthesis of bioactive polysaccharides and other metabolites, as well as for enhancing taxonomic identification. However, the number of sequenced genomes in this genus remains limited, hindering comprehensive research in these areas [35,36]. Chromosome conformation capture techniques such as Hi-C have been extensively utilized to significantly enhance the quality of assembled eukaryotic genomes [37], as well as of the microbial genomes from metagenomes [38,39]. However, their utilization in fungal genome research has been relatively scarce, with only one Hi-C-aided *Ganoderma* genome assembly (*G. boninense*) published to date [40]. In this study, we obtained a genome assembly for the strain *G. lucidum* 5.1 by employing a combination of high-throughput shotgun sequencing and the Hi-C method and explored its potential for xylomannan and fucogalactan synthesis.

## 2. Materials and Methods

### 2.1. Sampling Information

The *G. lucidum* strain 5.1 used in this study was obtained from the collection of the Laboratory of Biologically Active Compounds Biosynthesis at the Gause Institute of New Antibiotics. The ITS1 and ITS2 sequences of this strain have been deposited in GenBank under the accession numbers KT943370 and KT223755, respectively.

For long-term storage, the strain was maintained on potato-glucose agar at 4 °C. To initiate submerged cultivation, a nutrient medium was prepared, composed of the following ingredients per liter of purified water: 20.0 g of glucose, 10.0 g of peptone, 2.5 g of potassium hydrophosphate, and 0.25 g of magnesium sulfate. Submerged cultivation of the basidiomycete was performed in 750-mL Erlenmeyer flasks containing 100 mL of the liquid medium. The flasks were placed on an orbital shaker and agitated at 220 rpm at a temperature of 26 °C. To initiate the culture, mycelial agar plugs (3 mm in diameter) from seven-day-old agar cultures of the basidiomycete were used as inoculum in the seed liquid medium. The liquid seed mycelium was allowed to grow for 3 days. The amount of liquid inoculum was 10%. Submerged mycelium for further study was cultivated within 6 days. At the end of the cultivation period, the mycelium was separated from the culture liquid by filtration through a lavsan filter, followed by two washes with distilled water. Excess water was removed using filter paper, and the resulting raw mycelium was utilized for further experiments.

### 2.2. Sample Preparation and Sequencing

For the preparation of WGS libraries, DNA isolation from *G. lucidum* submerged mycelium was carried out using the DNeasy PowerLyzer Microbial Kit (Qiagen, Venlo, The Netherlands). Using the KAPA HyperPlus kits (Kapa Biosystems, Wilmington, MA, USA), two whole genome sequencing libraries were prepared—in two different sizes, with an average size of 480 bp and 700 bp, respectively. High-throughput sequencing was performed on the Illumina MiSeq (Illumina, San Diego, CA, USA) platform in paired 150-bp read format using the MiSeq Reagent Kit v2 (500 cycles) (Illumina, San Diego, CA, USA).

The preparation of Hi-C libraries included multiple stages [41]. First, the material was fixed with formaldehyde, which is necessary for stabilizing the configuration of the chromatin network. Then, nuclei were isolated by treatment with both ionic (0.3% SDS) and nonionic (1% Triton X-100, 0.5% NP-40) detergents, and most of the unfixed protein was removed to increase the accessibility of the genomic DNA for the enzymes used in subsequent stages of the protocol. The isolated nuclei were then treated with a frequent-cutting restriction endonuclease (DpnII) to cleave the genomic DNA into fragments of approximately 100–1000 bp in length. After chromatin fragmentation, the nuclei were incubated in the presence of a large fragment of *E. coli* DNA polymerase I (Klenow fragment; New England Biolabs, Ipswich, MA, USA), and all four deoxyribonucleotide triphosphates, one of which was conjugated with biotin (biotin-14-dATP). At this stage, the 5′-overhangs of the restriction fragments were filled in and labeled with biotin. The nuclei were then incubated in the presence of T4 DNA ligase (New England Biolabs, Ipswich, MA, USA), during which the restriction fragments that were in close proximity to the nucleus were ligated together (proximity ligation principle). In the next stage, the DNA was purified and sonicated, and the ligation products were selected on streptavidin-coated magnetic beads to prepare a library for deep sequencing on the Illumina platform using paired-end 150 bp reads.

### 2.3. Analysis of the WGS Data: Pre-Processing, Assembly and Quality Assessment

The preprocessing of sequencing data were performed by filtering based on hardware quality scores and by searching for and trimming adapter sequences using the Platanus_trim program v. 1.0.7 [42]. Whole-genome sequencing assembly was carried out independently using several software packages: Platanus_allee v. 2.0.2 [43], Platanus v. 1.2.4 [42], and SPAdes v. 3.11.1 [44]. The quality of the assemblies was evaluated using the BUSCO v. 5.2.2 program [45]. The assembly statistics were evaluated using the Quast package v. 5.0.2 [46]. The degree of microbial contamination was assessed using the Kaiju web service v. 1.9.0 [47] on the assembled contigs.

### 2.4. Total De Novo and Homology-Based Gene Annotation

Prior to the de novo annotation, to optimize the assembly obtained from the genome sequencing scaffolds, a preprocessing step was performed using the Redundans package v. 1.0.1 [48]. Subsequently, only the scaffolds longer than 1000 bp were selected for further annotation, resulting in a total of 1392 scaffolds.

For the identification and annotation of homologous genes, the GeMoMa software v. 1.9 for homology modeling was employed [49]. To perform de novo genome annotation, the FunGAP v. 1.0.1 software, specifically designed for annotating fungal genomes, was used [50]. FunGAP incorporates RNA-Seq data, which can be obtained from a closely related organism, to improve the accuracy of gene annotation. In this study, previously published RNA-Seq data from another strain of *G. lucidum* [51] were utilized for this purpose. The annotation pipeline for the genome is shown in Figure 1.

To assign functional annotations to the identified genes, the PANNZER2 service [52] was employed.

Additionally, a search for secondary metabolite biosynthetic gene clusters in the genome was conducted using the antiSMASH fungal version v. 7.0 [53] (strictness: “relaxed”).

### 2.5. Identification of the Genes Potentially Involved in the Synthesis of Xylomannan and Fucogalactan

We conducted a search for genes that potentially play a role in the synthesis of xylomannan and fucogalactan, specifically targeting *Och1p*, *Van1p*, *Anp1p*, *Mnn9p*, *Mnn2p*, and *Mnn6p* [31]. This genome-wide search encompassed the following sequential steps: (1) performing a BLAST search against the reference genome, (2) identifying genes whose GO profiles of transcripts corresponded to the desired functions using Euclidean distance based on a binary matrix of GO genes, (3) identifying the most closely characterized homologs of these transcripts through BLAST analysis, (4) characterizing the identified transcripts in terms of their enzymatic reactions associated with polysaccharide formation, leveraging the GO annotations and linking them to the Kyoto Encyclopedia of Genes and Genomes (KEGG), and (5) further exploring the KEGG database to identify metabolic pathways involved in polysaccharide synthesis that could potentially involve these proteins.

### 2.6. Analysis of the Hi-C Data

The initial processing of reads, mapping them to the genome assembly, and generating a chromosome contact density map were conducted using the Juicer package v. 1.6 [54] with default settings. To enhance the quality of the assembly and arrange scaffolds based on the contact map, we employed the 3D-DNA package [55] and Juicebox Assembly Tools v. 2.20.0 [56] for semi-automated scaffold reordering.

## 3. Results

### 3.1. Genome Assembly of Ganoderma Lucidum Strain 5.1

The WGS produced a total of 28.1 million raw reads, amounting to 4.2 billion base pairs. After filtering, 26.6 million reads remained for further analysis. Among the three assemblies evaluated (Platanus_allee, Platanus, and SPAdes), the Platanus_allee assembly demonstrated the highest quality based on the BUSCO assessment. Out of the 4464 orthologous gene groups, 4181 were identified in the Platanus_allee assembly, indicating a high level of assembly completeness. Therefore, this assembly was selected for subsequent analysis. Table 1 presents the primary quality evaluation of the chosen assembly.

The total length of the assembled genome is consistent with the expected size for this particular species of fungus and comparable to previous studies on the *G. lucidum* genome (e.g., 39.9 Mbp reported by [57]). The high N50 value indicates a high level of completeness in the assembly. Additionally, the analysis conducted using Kaiju revealed low contamination levels, with <0.2% of contigs identified as microbial.

Following the Hi-C sequencing aimed at improving assembly quality and merging scaffolds into pseudo-chromosomes (as described in the subsequent section), a total of 227 million read pairs were obtained. 

By mapping the Hi-C reads to the WGS assembly and subsequently reordering and combining the scaffolds, significant improvements were achieved in terms of N50 values and scaffold lengths (refer to Table 1 for details). The resulting debris—small, short contigs that could not be integrated into the assembly and contributed to an increased number of scaffolds—was subsequently eliminated.

### 3.2. Homology-Based Search of the Genes Involved in Xylomannan and Fucogalactan Biosynthesis

To validate the effectiveness of our gene search scheme, we conducted a test using the publicly available genome of a closely related *Ganoderma* species, specifically *Ganoderma* sp. 10597 [36]. The target genes of interest were successfully identified in this genome, and their list was further corroborated through a literature search (refer to Appendix A). These results provided strong evidence that our gene search scheme would also be applicable to our newly obtained genome. As expected, the subsequent search conducted on our own genome assembly yielded all 19 genes that were initially identified during the preliminary analysis (Table 2).

### 3.3. Search of the Genes Involved in Xylomannan and Fucogalactan Biosynthesis Based on the De Novo Annotation Results in WGS Assembly

The total annotation of the new strain’s genome revealed a total of 13,287 genes. A comparison between the de novo and homology-derived gene sets using the GFF-compare tool showed that 10,456 genes directly corresponded with the homology-derived genes. However, functional annotation was available for only 5540 of these genes in the reference genome (where a total of 6139 proteins were functionally annotated). To annotate the remaining 2831 genes without direct correspondence, functional annotation was performed using Pannzer2. Among this set, 710 genes obtained GO annotations, from which genes associated with polysaccharide metabolism were selected. A total of 44 genes related to polysaccharide metabolism were identified (the corresponding GO terms and descriptions can be found in Appendix A). The relevant GO terms and their descriptions are listed in Appendix A as a dictionary. A total of 375 genes in the annotated genome were identified as having homologs in the reference genome and being classified as participants in the complex polysaccharide metabolism process according to KEGG classification. The protein sequences of the transcripts for these genes are provided in Appendix A, significantly expanding the initial set of 19 genes.

### 3.4. Improving G. lucidum Genome Assembly Using the Chromosome Conformation Data

After Hi-C read mapping onto the assembly, generating a draft contact map, and applying automatic reordering of scaffolds in the genome assembly followed by manual correction, more than 85% of the initial genome assembly’s scaffolds were successfully grouped into 12 super scaffolds, which are presumed to correspond to the 12 chromosomes of the fungus (Figure 2). This observation aligns with previous findings on the number of chromosomes (12–13) in other *Ganoderma* species [40,58,59]. The contact map obtained after reordering, its corresponding labeling, and a table containing the coordinates and sizes of the super scaffolds can be found in Appendix A, respectively. Consequently, the incorporation of Hi-C data significantly enhanced the genome assembly and facilitated the precise localization of the genes of interest.

### 3.5. Prediction of Biosynthesis Genes Extends the Potential of Bioactive Compounds Production

To enhance our comprehension of the biosynthetic capabilities of *G. lucidum* beyond our initial focus on polysaccharides, we conducted a search for secondary metabolite biosynthesis genes within the genome. Twenty clusters of genes associated with secondary metabolite synthesis were identified on quasichromosomes 1, 2, 5, 8, and 11 (Appendix A). While most findings were related to terpene synthesis and the non-ribosomally synthesized peptides, a gene cluster on quasichromosome 11 was highly similar to the one responsible for the synthesis of clavaric acid (Figure 3). Notably, clavaric acid, a triterpenoid, is recognized as an inhibitor of human farnesyl-protein transferase, showcasing potent antitumor properties [60]; the genes responsible for its synthesis have recently been discovered in various *Ganoderma* species [61]. Our findings indicate the potential augmentation of the species’ anticancer properties conferred by the polysaccharides through the involvement of secondary metabolites.

## 4. Discussion

Mushrooms play a crucial role in planetary ecosystems, and their potential therapeutic applications have been recognized by national pharmacopeias. Advances in molecular genetics have enabled comprehensive analysis of mushroom functions, offering opportunities for the development of promising therapeutics to be explored in pre-clinical and clinical studies. Furthermore, these molecular approaches allow for a critical reassessment and refinement of the evidence from centuries of human use of bioactive fungi.

The *G. lucidum* genome presented in this study represents one of the first high-quality genomes generated for this species. It serves as a valuable resource for cataloging the genomic diversity of mushrooms, improving taxonomic profiling, and serving as a reference for multi-omics surveys. Further investigations utilizing various technologies, such as transcriptomics and untargeted metabolomics, will let this genomic potential be realized.

Our comprehensive approach to searching for polysaccharide synthesis genes has led to the identification of numerous genes potentially involved in their production. The abundance and diversity of these genes reflect the remarkable ability of the strain to produce xylomannan and fucogalactan. However, it should be noted that our survey did not reconstruct the entire biosynthetic pathway. The complexity of the carbohydrate skeleton and the expected intricate interactions between these genes pose challenges in deciphering their functional relationships. Conducting direct validation experiments, such as knockdown or knockout studies of specific genes, is challenging due to the limited genetic tractability of fungi. Validating these findings within a prokaryotic gene expression system presents challenges due to the significant evolutionary gap between prokaryotes and eukaryotes. Substantial disparities exist across various functional levels, notably in gene expression mechanisms and regulation; therefore, the potential for straightforward extrapolation is diminished. In contrast, utilizing a yeast-based system is a more favorable choice for subsequent validation efforts (though complexities can arise from differences such as cellular structure).

In addition to medicinal perspectives, our findings also have biotechnological and ecological implications. For instance, our earlier work demonstrated that the mycelium extract of the same strain 5.1 can be utilized as a sensitive coating for gas acoustic sensors [62], highlighting the potential role of complex carbohydrates in such physical properties. Moreover, recent studies have shown the use of *Ganoderma* for producing biodegradable substrate materials for sustainable electronics [63]. These applications highlight the broader relevance of *Ganoderma* research beyond medicine.

Our study is also among the first to employ Hi-C sequencing in fungal surveys, resulting in high-quality contact maps and improved genome assembly. This approach has only recently been applied to edible and medicinal mushrooms [64]. The substantial improvement in assembly quality suggests that, following the spirit of the Earth Biogenome Project [65], outlining the genomic diversity of fungi can be efficiently achieved using Hi-C sequencing, following a methodological platform such as DNA Zoo (http://www.dnazoo.org, (accessed on 10 November 2022)), for the cost-effective generation of high-quality fungal genomes. Of note, Hi-C allows for the identification and repair of misassemblies via semi-manual visual inspection. Furthermore, if the fungal sample has an associated bacterial consortium (e.g., microbiome) of sufficiently high abundance, then improved assemblies can be obtained for the respective microorganisms.

There are certain limitations to the method that should be mentioned. First, there is the necessity of preparing a Hi-C library (in addition to the WGS library), which incurs extra costs. Moreover, the Hi-C protocol itself entails multiple steps that are optimized for better scalability. There are a couple of experimental challenges specific to fungi that need to be addressed. Firstly, the presence of a cell wall complicates the cell lysis step, necessitating additional mechanical destruction. At this step, careful temperature control is crucial to prevent DNA fragmentation by endogenous nucleases. Secondly, the high protein/DNA content of fungi hinders fermentative reactions. Therefore, accurate removal of cell debris is essential before incubating chromatin with the restriction enzyme. Furthermore, the success of applying Hi-C strongly depends on the quality of the WGS assembly itself: if the original contigs are short, noisy, or unresolved at the highly-repetitive genomic regions, then even with the high coverage, the benefits of Hi-C reads will be inferior. Therefore, replacing Illumina WGS with WGS using long-read sequencing can significantly improve this situation, as well as aid in genome phasing when the analyzed fungal sample is diploid.

While our study did not specifically focus on analyzing chromatin patterns, a preliminary examination of the chromosome contact map revealed intriguing original structures that warrant further investigation, particularly in combination with gene expression profiling and under different experimental conditions. Advancing 3D fungal genomics and understanding the links between functional and structural organization will contribute to a comprehensive understanding of the fungal kingdom’s biology.

## 5. Conclusions

Our study sequenced one of the first *Ganoderma* genomes using Hi-C-augmented sequencing and identified key genes involved in polysaccharide synthesis. This genome assembly provides a valuable resource for comparative genomics and multi-omics profiling. Future analyses will explore the genome’s spatial organization and its implications for biological processes. Our findings shed light on *G. lucidum* genomic diversity and contribute to the basis for its applications in medicine and biotechnology.

## Figures and Tables

**Figure 1 jof-09-01020-f001:**
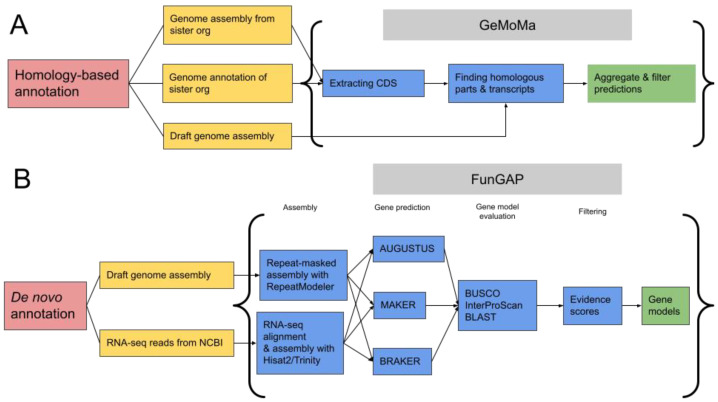
Gene annotation workflows. (**A**) Homology-based. (**B**) De novo.

**Figure 2 jof-09-01020-f002:**
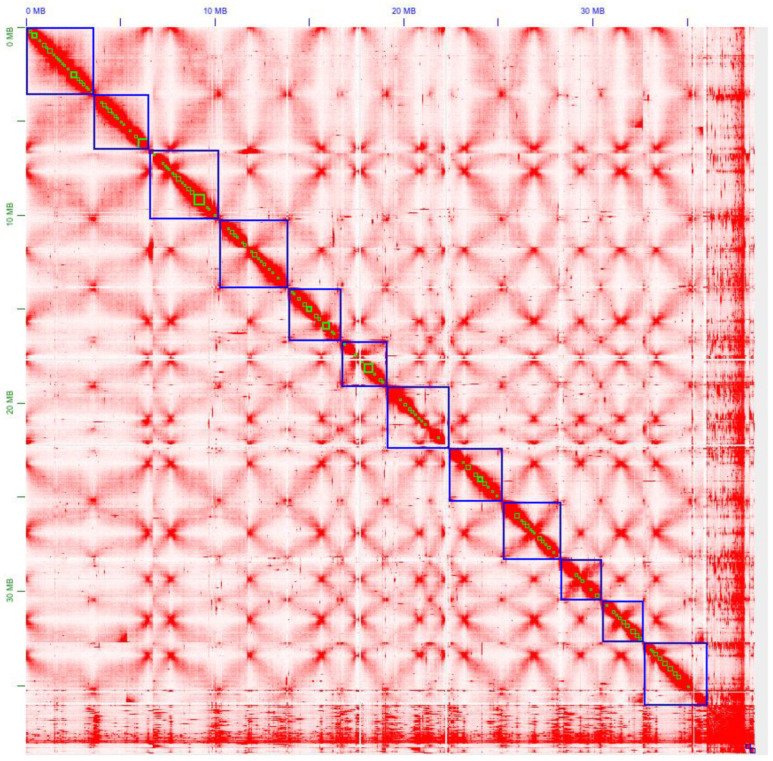
Chromosome contact intensity map of the *G. lucidum* genome following the reordering of scaffolds. Resolution: 100 kbp. Super-scaffolds/quasi-chromosomes derived from Hi-C data are depicted in blue, while the largest scaffolds from the original WGS assembly are illustrated in green.

**Figure 3 jof-09-01020-f003:**
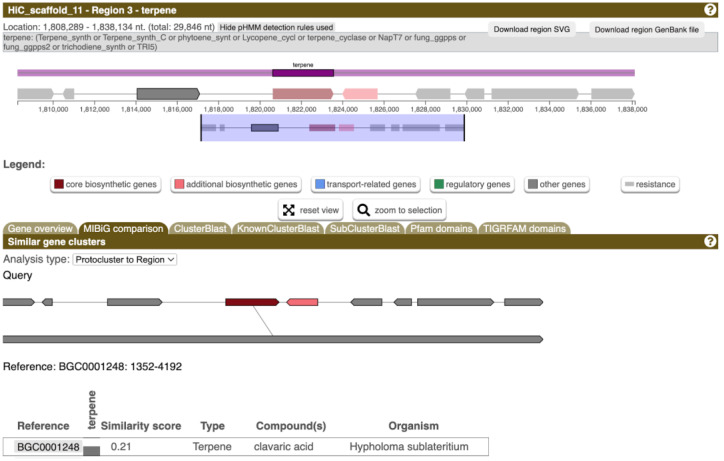
Clavaric acid biosynthesis gene cluster according to the antiSMASH prediction.

**Table 1 jof-09-01020-t001:** Genome assembly statistics.

WGS + Hi-C Assembly	WGS Assembly	
38,961,136	39,106,447	Total assembly length, bp
2731	2270	Number of contigs
3,048,500	72,187	Scaffold N50, bp
4,258,832	594,110	Maximum scaffold length
55.61	55.57	GC%

**Table 2 jof-09-01020-t002:** Genes potentially involved in xylomannan and fucogalactan synthesis were identified in the *G. lucidum* strain 5.1 genome. The e-values were 0 for all entries.

AA Sequence Identity, %	Gene ID in *G. lucidum* Strain 5.1 Assembly	Reference Gene ID in *Ganoderma* sp. 10597
100	MRNA_403_R0	jgi|Gansp1|38124|e_gw1.1.2377.1
99.42	MRNA_2449_R0	jgi|Gansp1|41514|e_gw1.2.2688.1
99.74	MRNA_1789_R0	jgi|Gansp1|42603|e_gw1.2.2497.1
98.33	MRNA_7121_R0	jgi|Gansp1|51293|e_gw1.7.902.1
99.13	MRNA_7035_R0	jgi|Gansp1|51570|e_gw1.7.313.1
98.97	MRNA_10444_R0	jgi|Gansp1|55798|e_gw1.11.587.1
93.11	MRNA_10444_R1	jgi|Gansp1|56496|e_gw1.11.376.1
99.74	MRNA_6022_R0	jgi|Gansp1|69736|estExt_Genewise1.C_6_t10369
99.48	MRNA_8855_R0	jgi|Gansp1|73716|estExt_Genewise1.C_9_t20352
99.01	MRNA_4151_R0	jgi|Gansp1|84501|estExt_Genewise1Plus.C_4_t10328
96.12	MRNA_6859_R0	jgi|Gansp1|88381|estExt_Genewise1Plus.C_7_t10263
99.27	MRNA_7000_R0	jgi|Gansp1|88625|estExt_Genewise1Plus.C_7_t20055
98.83	MRNA_4436_R0	jgi|Gansp1|127772|fgenesh1_pm.4_#_322
100	MRNA_7163_R0	jgi|Gansp1|129696|fgenesh1_pm.7_#_370
97.8	MRNA_11372_R0	jgi|Gansp1|132299|fgenesh1_pm.13_#_149
99.86	MRNA_1100_R0	jgi|Gansp1|113974|fgenesh1_kg.1_#_1092_#_isotig08803
99.59	MRNA_2242_R0	jgi|Gansp1|115010|fgenesh1_kg.2_#_583_#_isotig10382
99.74	MRNA_6364_R0	jgi|Gansp1|118777|fgenesh1_kg.6_#_527_#_isotig08567
99.85	MRNA_7916_R0	jgi|Gansp1|120249|fgenesh1_kg.8_#_247_#_isotig07580
97.78	MRNA_11130_R0	jgi|Gansp1|123003|fgenesh1_kg.12_#_593_#_isotig07483

## Data Availability

Sequencing data and genome assembly have been deposited at the ENA under accessions PRJEB63142 and ERZ21549728, respectively.

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
