# Peer review of "Unraveling the Polysaccharide Biosynthesis Potential of *Ganoderma lucidum*: A Chromosome-Level Assembly Using Hi-C Sequencing"

_jof, 2023, doi:10.3390/jof9101020_

Round 1
Reviewer 1 Report
The polysaccharides of Ganoderma lucidum is an important active component with physiological functions such as anti-tumor. However, the biosynthetic pathway of Ganoderma lucidum polysaccharides has not been elucidated yet. The research results of this article provide a theoretical basis for the biosynthesis of Ganoderma lucidum polysaccharides and other mushroom polysaccharides, and have great scientific significance and application value. However, the manuscript only obtained synthesis enzyme genes that may be related to the polysaccharide synthesis pathway from the genome of Ganoderma lucidum through bioinformatics analysis, and did not further identify the functions of these genes. Therefore, it is not possible to determine which genes constitute the polysaccharide synthesis pathway of Ganoderma lucidum. Therefore, it is recommended that the author use a prokaryotic gene expression system or a yeast gene expression system to perform functional identification of the predicted genes and analyze the synthesis pathway of Ganoderma lucidum polysaccharides.
In addition, the "Ganoderma lucidum" shoule be Italic, and have some not Italic in the manuscript such as line 137.
Author Response
Greetings! This is an updated answer of the review.
Dear Editors and Reviewer,
We appreciate your remarks and questions. We addressed them in a point-wise manner, Please find the replies below and the amended resubmitted text .
In addition to updating the meaning of the text, we have also performed additional language editing where required and actualized the information about the funding and affiliations.
The data have been deposited, and the data availability section was updated accordingly.
Additionally, we would like to request that the corresponding author is changed from Ignat Sonets to Larissa Krasnopolskaya.
With best regards,
on behalf of the authors,
Ignat Sonets
Reviewer 1
The polysaccharides of Ganoderma lucidum are an important active component with physiological functions such as anti-tumor. However, the biosynthetic pathway of Ganoderma lucidum polysaccharides has not been elucidated yet. The research results of this article provide a theoretical basis for the biosynthesis of Ganoderma lucidum polysaccharides and other mushroom polysaccharides, and have great scientific significance and application value. However, the manuscript only obtained synthesis enzyme genes that may be related to the polysaccharide synthesis pathway from the genome of Ganoderma lucidum through bioinformatics analysis, and did not further identify the functions of these genes. Therefore, it is not possible to determine which genes constitute the polysaccharide synthesis pathway of Ganoderma lucidum. Therefore, it is recommended that the author use a prokaryotic gene expression system or a yeast gene expression system to perform functional identification of the predicted genes and analyze the synthesis pathway of Ganoderma lucidum polysaccharides.
We appreciate your proposal. Indeed, an experimental verification of the bioinformatically derived in silico predictons would be a valuable addition to our study. While lack of such confirmation can be be viewed as a certain limitation of the present work, we suggest this is out of scope of the currently presented stage of our research.
Noteworthy, regarding the prokaryotes, they are evolutionary very distant from eukaryotes and there are huge differences in their functioning at different levels, particularly, the mechanisms of gene expression and its regulation, thus reducing the evidence of simple extrapolation based on a prokaryotic system. Compared to that, a yeast-based system offers more advantages compared to a prokaryotic system and validation can be conducted in subsequent work ( though can still be complicated by the differences, e.g. in cellular structure). We have added these considerations to the Discussion part of the manuscript.
Provided limited opportunities to extend the experimental part of our work, to increase the value of the presented data we extended our in silico analysis - by performing the biosynthesis genes search in the G. lucidum genome that further revealed additional metabolic functions potentially involved in the mushroom’s bioactivities. Their description has been added to the Results section.
In addition, the "Ganoderma lucidum" shoule be Italic, and have some not Italic in the manuscript such as line 137.
Thank you for this remark. We have updated the formatting, accordingly.
Reviewer 2 Report
1. The paper explores genes associated with xyloglucan biosynthesis; does this strain also produce other types of polysaccharides? What are the genes associated with the synthesis of other types of polysaccharides?
2. The paper focuses on one strain, Ganoderma lucidum strain 5.1, is this strain used for production?
This strain has " a remarkable producer of xylomannan ", what is the specific yield?
3. Can the Hi-C-assisted genome assembly used in the paper be applied to other fungal genomic studies? What are the advantages and limitations of this technique in sequencing fungal genomes?
The english language should be improved.
Author Response
Greetings! This is an updated answer of the review.
Dear Editors and Reviewer,
We appreciate your remarks and questions. We addressed them in a point-wise manner, Please find the replies below and the amended resubmitted text.
In addition to updating the meaning of the text, we have also performed additional language editing where required and actualized the information about the funding and affiliations.
The data have been deposited, and the data availability section was updated accordingly.
Additionally, we would like to request that the corresponding author is changed from Ignat Sonets to Larissa Krasnopolskaya.
With best regards,
on behalf of the authors,
Ignat Sonets
- The paper explores genes associated with xyloglucan biosynthesis; does this strain also produce other types of polysaccharides? What are the genes associated with the synthesis of other types of polysaccharides?
(We suppose the reviewer might have been referring to xylomannan, as xyloglucan is another polysaccharide with different structure.) The chemical diversity of Ganoderma lucidum-produced polysaccharides is known. From the strain 5.1, previously we have isolated novel polysaccharides - xylomannan and fucogalactan - and established their structures. In the present work we identified the genes potentially associated with biosynthesis of these two compounds based on existing knowledge (without separating the genes into two compound-specific groups). The search for the genes involved in synthesis of other polysaccharides was not the aim of the study. To improve clarity, we have amended the text accordingly.
- The paper focuses on one strain, Ganoderma lucidum strain 5.1, is this strain used for production?
Despite its prominent capabilities (see below), currently the strain is not used in an industrial-scale production so far.
This strain has " a remarkable producer of xylomannan ", what is the specific yield?
The yield of xylomannan is 600-700 mg/l.The content of xylomannan in submerged mycelium is 2.6%. We have added this information, as well as similar data about the second investigated polysaccharide - fucogalactan - to the text.
- Can the Hi-C-assisted genome assembly used in the paper be applied to other fungal genomic studies? What are the advantages and limitations of this technique in sequencing fungal genomes?
Yes, the technique is definitely universally applicable for investigating fungal genomes, and it has recently started being used for a variety of edible and medicinal mushrooms, as we had also briefly mentioned in the Discussion section. Our case study is an example of how the technique commonly employed for studying other eukaryotic clades proves also useful for fungi.
Advantages:
- Drastic increase of assembly quality metrics.
- Chromosome-level scaffolding.
- Ability to discover and repair misassemblies via semi-manual visual inspection.
- If the fungal sample has an associated bacterial consortium (microbiome) of sufficiently high abundance, then improved assemblies can be otained for the respective microorganisms.
- As a extra bonus, 3D genome of the fungus with chromatin patterns is revealed.
Disadvantages:
- Need to prepare an additional library which is costly. Protocol requires multiple steps and is yet to be scaled.
- There are a couple of experimental challenges specific to fungi that need to be addressed. Firstly, the presence of a cell wall complicates the cell lysis step, necessitating additional mechanical destruction. At this step, careful temperature control is crucial to prevent DNA fragmentation by endogenous nucleases. Secondly, the high protein/DNA in fungi hinders fermentative reactions. Therefore, accurate removal of cell debris is essential before incubating chromatin with the restriction enzyme.
- The success of applying Hi-C strongly depends on the quality of the WGS assembly itself: if the original contigs are short, noisy or unresolved at the highly-repetitive genomic regions then even with the high-coverage the benefits of Hi-C reads will be inferior. Therefore, replacing Illumina WGS with WGS using long-reads sequencing can significantly improve this situation. As well as aid in genome phasing when the analyzed fungal sample is diploid.
To further address your request, we have amended the discussion section by adding a few facts about the Hi-C-assisted genome assembly.
Comments on the Quality of English Language: The english language should be improved.
Thank you, we have checked the text and performed additional language editing.